# The Breath of the Metropolis: Smart Working and New Urban Geographies

**Fulvio Adobati** [1,*] and **Andrea Debernardi** [2]

1 Centre of Territorial Studies, Department of Engineering and Applied Sciences, University of Bergamo, I-24129 Bergamo, Italy
2 META srl, I-20900 Monza, Italy; andrea.debernardi@metaplanning.it
* Correspondence: fulvio.adobati@unibg.it

**Abstract:** The paper explores the potentialities of telework, a topic with rich literature published since the 1970s, which has become topical again with its forced application related to the COVID-19 pandemic emergency. The paper carries out an analysis of the potential territorial impact—transport networks and geographies of living—of telework in the Italian national context. The analysis highlights the potential relevance of the application of telework in certain metropolitan areas that present urban poles where economic sectors with a high propensity for telework are centralised. This survey relates the large stock of tourist housing in the vicinity of large metropolitan areas to a potential demand arising from the change in housing preferences towards more pleasant contexts made possible by the application of telework. In conclusion, this work aims to contribute to the construction of a platform for the Italian context—lagging behind but with recent legislative measures on smart working—aimed at favouring the definition of research lines able to enhance the potential offered by the application of telework for environmental, social, and territorial sustainability objectives, and it also aims to outline possible territorial scenarios for the main metropolitan areas

**Keywords:** smart working; transport system; mobility demand; housing stock; urban geographies; working from home (WFH)

## 1. Introduction

One of the most important effects of the COVID-19 pandemic crisis has been a deep change in travel behaviour. The need for strict interpersonal spacing and the fear of infection have led to unprecedented levels of adoption of remote online activities, including smart working [1–8].

Facing this unexpected situation, several authors have argued for a strong change in social attitudes towards remote jobs, or e-work, in the next years. For instance, Tremblay [9] notes that periods of crisis often generate radical transformations, suggesting that they could modify the whole organisation of work at the metropolitan and regional level. Following this point of view, the forced reaction to pandemic constraints shows that remote working can be, in many cases, a good organisational solution, generating personal, economic, and environmental benefits without any loss of job productivity. Such evidence contributes to weakening a number of traditional resistances, acting in favour of a high level of remote working even in the post-pandemic condition.

Actually, the theme of smart working is not completely new for planning researchers: it indeed plunges its roots into the wide interest that arose in telework and telecommuting in the 1970s and has grown up during the Information and Communication Technologies (ICTs) revolution up to the first decades of the twenty-first century. Fifty years of scientific literature highlight a number of aspects of remote activities, impacting directly or indirectly on job organisation, human attitudes, mobility demand, and even settling structures. Taken as a whole, these impacts can generate very complex and tendentially ubiquitous but often highly selective processes.

This contribution aims to explore the dense network of possible spatial effects produced by the potential consolidation of smart working in Italy, with particular reference to the main metropolitan areas. In fact, in the main metropolitan regions, due to the number of activities suitable for telework and the presence of a substantial under-utilised building stock (semi-abandoned villages combined with significant stocks of tourist housing), located in surrounding areas characterised by high levels of marginality.

The paper opens with a review of the literature devoted to telework (par. 2) and its potential socio-economic, geographical, and lifestyle implications, with reference to the experiments and experiences that have been conducted. This is followed by a section devoted to the methodological approach adopted, aimed at outlining the direct impacts of the application of telework and the correlative rebound effect (par. 3), and building a hypothesis for the empirical analysis carried out in the Italian context (par. 4), where, with the approval of new guidelines by the national government on contractual labour relations in both the public and private sectors, a consolidation of smart working is underway. The approach adopted first examines the direct impact on the demand for mobility with the potential effects on the transport networks, followed by an evaluation of the possible rebound effects exerted by the opportunities offered by the loosening of the constraints on residence offered by the adoption of telework, correlating to a dynamic of modification of settlement preferences with the stock of second homes located in areas peripheral to the metropolitan polarities.

## 2. Telework, a Quite Long History (and a Boundless Bibliography)

The term "telecommuting" was coined quite accidentally by Nilles during the energy crisis of 1973 [10], and in a few years, a focus was opened on the possible trade-off between telework and commuting [11]. During the 1970s, very optimistic forecasts were made about the possible diffusion of telework: for example, AT&T predicted that in 1990 all the US workforce would be working from home [12], and even a decade later, it still predicted that the share of telecommuters would reach half of the total [13]. Quite similarly, in the early 1980s, the Institute for Future Studies foresaw that 40% of the US employment would be telecommuting by 2000.

From the very beginning, the concept of telework was strictly connected to the development of ICTs, considered as fundamental tools allowing employees to work remotely from their office, regardless of whether the remote workplace was home-based or not. Therefore, already in the 1980s, increasing attention was devoted to the impact of ICTs on travel time and commuting. Salomon [14–17] explored the relationships between telecommunications and transport, showing that information exchanges can substitute for physical travels but also generate the need for new journeys with complementarity effects, the final outcome being uncertain.

In the two last decades of the twentieth century, the pervasive character of work issues attracted the attention of many disciplines, such as business economics, work sociology, town planning, and transport engineering, leading to a wide set of studies about telecommuters and their features [18–22]. This rapidly led to a quite boundless research field, split into many segments, each corresponding to a single viewpoint on the theme.

A first large-scale pilot project was developed by the State of California, with the PATH program, which was specifically addressed to public sector employees [23–28]. This project turned out to be useful for deepening the knowledge of driving processes leading people to telecommute, but it did not result in clear outcomes about its potential for reducing mobility. Further studies were carried out during the 1990s from several points of view [29–35], and in pandemic times it can be interesting to remember a specific deepening concerning the sudden growth of telework in the day immediately after the earthquake of San Francisco in 1991, and its structural effects on commuters' behaviour [36].

Nevertheless, none of these researches found any particular evidence of a boom of remote working. At the turn of the new century, it became clear that the optimistic predictions of the 1970s and 1980s had not come to pass, with empirical evidence indicating the

existence of strong barriers to telework among both companies and employees. Actually, companies perceive high costs of implementing remote working programs in terms of organisational change, technological investments, reduced control, and possible productivity losses, while employees do not necessarily feel that telecommuting is a way to improve their work–family life balance [37], sometimes fearing a reduction of social interaction. At the same time, relationships between telework and physical commuting prove to be more complex than they seem.

As a consequence of this relative disappointment, in more recent years, a paradoxical situation has come to be, with a stagnant growth rate in remote working experiences and a general loss of interest in telecommuting as a specific feature appearing just simultaneously to the fastest development of the ICTs economy, which in turn tends to support a wide diversification of teleworking tools and practices.

One of the principal effects has been a tendency to widen the study field of remote working, including new approaches and practices [37]. If, during the last decades of the twentieth century, the focus was mainly on work performed by employees during paid hours in a fixed place (home or satellite office) different from the normal worksite and formally approved by the employer, with minor attention for self-employment, in more recent years studies encompass other categories, such as overtime teleworkers (home-based work performed outside working hours [38,39]) and mobile (or nomadic) workers, operating while they are travelling or in other places, using mobile ICT such as portable computers and mobile phones [34,40–45].

At the same time, the theme became more popular in other countries, with new studies based on different viewpoints and sensibilities, such as in Australia [13,46–64], Canada [39,43,65–79], and the UK [80–91].

European researchers approached the theme of what was called eWork mainly on the basis of a focus on ICT potentials [92–95], with less emphasis on the commuting impacts. This point of view became quite popular in Nordic countries and Baltic states [34,96–108].

The attention for the potential effects of remote working was even more timely in the Netherlands [109–125]; it was quite quickly followed by Belgian researchers [126–133].

An original viewpoint, often rooted in social theories, was proposed in France at the very first time of the debate [37,134–144] and in Switzerland [145,146].

The Spanish and Portuguese experience is more recent, mainly because it mirrors public efforts to enhance remote working by telecentres [147–152]. On the other hand, in the last fifteen years, remote working attracted great interest as a way to sustain the development of small Mediterranean islands, especially in Greece [153–162].

Finally, in the Italian case, researchers have devoted comparatively little attention to the theme of teleworking, with only a few studies, mainly funded by telecommunications firms or international entities [44,163–167]. However, these contributions have tended to highlight juridical or social aspects (e.g., in public employment), with only some focused on spatial effects of smart work centres and other solutions.

In short, on the eve of the pandemic crisis, teleworking could be regarded as a worldwide research theme in the field of ICTs, repeatedly invoked as a panacea for a wide range of societal problems, ranging from eternal economic growth to bridging social and cultural gaps via the internet [118]. According to this approach, many scholars and policymakers continued regarding it as a viable instrument to solve congestion problems and reduce the environmental impact of road traffic [102,113,168,169]; to increase workers' well-being by reducing travel-related stress, providing a better work environment and improving work–family life balance [69,170–172]; or to allow companies to save money through lower real estate costs and productivity gains [173]. On these bases, remote working schemes were continuously included in periodic promotional campaigns promising its imminent "take off", which were in turn reinforced by advances in remote communications technology and increasing environmental constraints [37].

Nevertheless, while nomadic work and overtime home-based telework were increasing [174], concrete remote working experiences in the original meaning remained relatively

limited. Telework seemed to be structurally caught in a low-equilibrium trap [37], and its diffusion had become something like a never-ending promise, its future always just around the next corner [175].

In this context, the arguments repeatedly used to glorify the improbable future of telecommuting seemed no longer convincing [37], unless to hypothesise an external shock, such as, indeed, a major epidemic [86].

At the same time, it is surprising that half a century of research has led to quite small results, with few advancements from a theoretical point of view, with the consequence that the boom of remote working induced by the COVID-19 crisis paradoxically catches scholars and policymakers in a condition of only partial readiness in understanding and managing a suggestive phenomenon, which is much more complex than it seems [6].

## 3. Teleworking Potential and Its Effects: Towards a Study Hypothesis for Italy

Despite of their long-term history, telework and telecommuting seem not to have a common definition [37,120,174] yet. Initially (from the 1970s to the 1990s), telework was fundamentally intended as home-based work, performed by workers during paid hours, without any reference to ICTs: from this perspective, it was considered a synonym of telecommuting. Later, the concept was extended to include any work performed outside the official workplace, be it a place different from home or even a means of transport (mobile or "nomadic" work), and the linkage with ICT was increasingly underlined.

Some of the most popular definitions of teleworking or telecommuting are reported below:

- using telecommunications technology to work from home, or at a location close to home, during regular work hours, instead of commuting to a conventional workplace at a conventional time [176];
- work carried out using ICT at a place other than that where the results of the work are needed (European Commission);
- work conducted from a location other than the conventional worksite whilst connected to the firm's computer systems by means of information and telecommunications technology (ICT) [37].

These definitions cover a wide set of different situations, in terms of location, intensity in time, contractual arrangement, used technology, and so on [120,177–180].

They include, for instance, home-based and telecentres-based remote workers, as well as mobile teleworkers, without any reference to the duration and frequency of working out of the office: therefore, working at home a day per week, in the evening, or during the weekends is now commonly considered teleworking.

Regarding workplaces, in 2003, the Statistical Indicators Benchmarking the Information Society (SIBIS), established by the European Commission, defined four different types of telework: telework from home, mobile telework, freelance telework in SOHOs (small office/home office), and telework done in shared facilities outside of organisations and home [1]. However, many definitions of remote working do not include self-employed work at home (artists, writers), but only workers who occasionally work at home, whether that is on a regular or formal basis or not.

Mokhtarian [176] suggested a clear distinction between home-based and non-home-based telework; where the first category included already several cases:

✓ running a home-based business as one's only job;
✓ running a secondary home-based business, in addition to holding another job;
✓ bringing overtime work home after a full day at the office;
✓ working at home rather than in the office.

On the other hand, non-home-based remote work was related to even more numerous situations:

✓ working from a satellite centre closer to home than the primary office;
✓ managing a branch office;

✓　performing some field activities outside the office;
✓　working while travelling.

Moreover, she pointed out a further form of remote work: long-distance commuting, which can be either home-based or non-home-based.

Frequency of teleworking is another key issue for its definition and typology. Actually, already Nilles [10] noted that "… most home-based telecommuting is (and is likely to be) part-time …"; and a continuous gradient can be established from few workers acting entirely from home to a wider set of people occasionally working outside their workplace [181]. A clear boundary between these categories is generally difficult to draw, because teleworking is often directly connected to flexibility in working time [145,182], and it is frequently developed on an informal basis [127,183], making measurement difficult even within organisations in which it is practised [181].

The International Labour Organization (ILO) suggest a classification based on three main types of telework, highly related to the use of ICT (see Table 1).

**Table 1.** Types of telework.

| Modality | Use of Technology | Location |
|:---:|:---:|:---:|
| Regular home-based telework | Always or almost all the time | From home at least several times a month and in other locations less often than several times a month |
| High mobile telework | Always or almost all the time | At least several times a week in at least two locations other than the employer's premises or working daily in at least one other location |
| Occasional telework | Always or almost all the time | Less frequently and/or fewer locations than high T/ICTM |

Source: Eurofound-ILO [184].

This typology tends to assume a correlation between workplace and frequency of teleworking, which seems not to be confirmed in real experiences. A different solution, proposed by Ravalet et al. [146], can be developed simply by crossing the location (home-based, other fixed (satellite centres, telecottages, etc.), mobile) and the frequency (regular, occasional, overtime, etc.), in order to obtain six main categories (see Table 2).

**Table 2.** Types of telework.

| Location | Frequency | |
| | Total | Partial |
|:---:|:---:|:---:|
| Home-based telework (HBT) | H | H′ |
| Satellite centre telework (SCT) | S | S′ |
| Nomadic/mobile telework (NMT) | N | N′ |

Source: Ravalet et al. [146].

These definitions can be used to develop some more specific study hypotheses about the future prospects of telework in the Italian case, and their effects on the urban and natural environment. From a theoretical viewpoint, it is possible to identify three different steps:

(A)　assessing the teleworking potential, i.e., the possible number of workers which could adopt remote working in a "normal" (not pandemic) situation;

(B)　estimating the direct effect of telecommuting increase, in terms of reduced car mileages and air pollution;

(C)　identifying possible indirect effects, linked to possible rebound situations on the location of work/residential activities.

Each of these levels has been widely discussed by the scientific literature at an international level, driving to some general results that can be assumed as study hypothesis on the less-known Italian case.

### 3.1. Teleworking Potential

Regardless of these classifications, the diffusion of remote working is highly dependent on its acceptability in terms of economic, organisational, and individual impacts.

Mokhtarian and Salomon [185] propose a framework of constraint and incentives for the adoption of telework from the worker's perspective. Constraints and disincentives include awareness about the possibility of telework and organisational rigidity, such as work monitoring or job characteristics. Personal disincentives include doubt about discipline in working productively at home, the perception of some utility in commuting [186,187], the willingness to interact with other workers and to be physically noticed, the desire to avoid risks for their career opportunities, and the lack of additional space at home. On the other hand, incentives include higher productivity with less disturbance from colleagues, time flexibility, the possibility of better combination between work and family/personal activities, reduction of commuting costs, and finally, ideological reasons supporting environmental sustainability (see Table 3).

**Table 3.** Factors influencing the decision to telework.

| | | Employers | | Workers |
|---|---|---|---|---|
| **Incentives** | ✓ | Higher productivity during hours worked | | |
| | ✓ | Reduced absenteeism | ✓ | Work-related (no disturbances) |
| | ✓ | Possibility of more hours worked per day where travel time is replaced by work time | ✓ | Family-related (more flexible, more time with family) |
| | ✓ | Improving of quality of work (reduced stress) | ✓ | Better work–life balance (particularly for women) |
| | ✓ | Reduced number of workspaces ("hot desking") with savings in office rentals and running costs | ✓ | Leisure-related (more time for self) |
| | ✓ | Widening of recruiting area for staff | ✓ | Travel-related (no commuting costs) |
| | ✓ | Weakening of control over quality of work and reduced chance to control employees | ✓ | Ideological (saving energy, sustainability) |
| **Disincentives** | ✓ | Need for organisational changes, especially for the management of the psycho-social distance from the work environment | ✓ | Lack of discipline |
| | | | ✓ | Utility from commuting |
| | | | ✓ | Psychosocial factors (sense of isolation) |
| | ✓ | Loss of benefits of interactions between workers | ✓ | Risk that work ends up invading family life |
| | ✓ | Risks for data security | ✓ | Risk constraints |
| | ✓ | Teleworking can be seen as "counter-cultural" to the ethos of organisation, as a result of reducing face-to-face contact and accessibility of staff to management | ✓ | Cost constraints |
| | | | ✓ | Additional costs in lighting and heating of the home and provision of work space. |

Sources: [9,32,37,67–69,72,73,76,99,118,170,171,181,185,188–206].

The adoption of teleworking can be analysed on the basis of the theoretical framework proposed by Baruch and Nicholson [190], which is based on a semi-structured survey of 62 subjects in five UK organisations, both public and private. Following this framework, the adoption of smart working is influenced by four principal factors (see Figure 1), related

to the job nature, to the organisational constraints, to the family/home situation, and to individual attitudes. Telework becomes feasible and effective only if all factors are fulfilled [1].

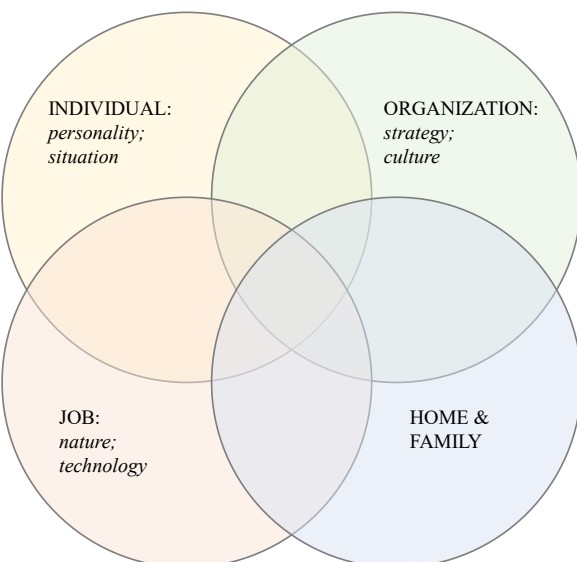

**Figure 1.** Main factors for telework adoption. Source: Baruch and Nicholson [190].

The internet-based survey developed by White et al. [86] on the staff of the UK Department for Transport found that barriers to the adoption of teleworking were linked mainly to the need of contact with colleagues, the inadequacy of IT facilities, and the lack of encouragement by the management. On the contrary, the strongest motivations for teleworking were the increase in productivity and/or quality of work, greater flexibility, more free time, reduced congestion, and transport costs. Aguilera et al. [37] argue that the biggest barriers to smart working are social, including a change in management practices and the need to review work organisation, as well as the expectation of productivity losses (which are often overestimated). Hynes and Rau [207] underline that remote working is often marginalised by businesses, especially when appropriate regulation and guidelines are lacking.

In their study about the Swiss situation, Ravalet et al. [146] highlight that the propensity to telework is higher for workers in small firms with high levels of responsibility and/or time flexibility in their job. They explain these results with the importance of personal trust and responsibility in remote working. This result is confirmed by many other studies [37,95,120,174,208] Consequently, smart working is often a matter of informal and highly individualised agreements, and its adoption rate is strongly variable between economic sectors and activities.

Social factors affecting telework choice in the ESA are specifically analysed by Walls et al. [209] and Sener and Bhat [210], who highlight significant correlations with individual and family indicators, such as age, gender, education, and the presence of children in the household. Some other studies find that women especially are inclined to telework [211].

At the same time, the importance of cultural factors should be not neglected. Comparing two locations of the same ICT multinational company in France and the Netherlands, Peters et al. [212,213] and Peters and Batenburg [214] shows how telework adoption among line managers can be affected by cultural factors: in the Dutch context, remote working was seen as a fully acceptable practice, whereas in France the assessment of power distance and uncertainty avoidance contributed to weakening it.

Nevertheless, the technological dimension also plays a role, for example, in terms of the availability of proper internet connections. For example, Halford [215] find that 80% of teleworkers use computers and telephones in their work.

Last but not least, the likelihood of adopting teleworking highly depends on the nature of jobs and their economic sector: in particular, a number of different jobs, needing face-to-face interaction (for example, medical treatment, hotels, catering, teaching), are clearly excluded from the possibility of remote working [86,181].

Following these elements, teleworking is the result of a very selective process, and the "Archetypical Teleworker" [118] typically belongs to very specific job categories, with a high level of flexibility and responsibility and good computer access. Fu et al. [216] point out that teleworkers are more likely to be higher professional workers, and many other scholars show that telecommuters are older, wealthier, and better educated than non-telecommuters [90,217,218].

Studies on the French and German situation [219] are even more sceptical, pointing out that remote working tends to represent a limited practice, essentially adopted by a few intellectual professions characterised by a considerable amount of job autonomy.

Analysing a sample of the 6th European Working Conditions Survey (more than 20.000 workers), López-Igual and Rodríguez-Modroño [220] confirm that remote workers are mainly high-skilled men living in urban areas. Nevertheless, their study underlines a growing heterogeneity, with a relevant percentage of technicians and clerical support workers, as well as of women (especially on home-based telework); following this evidence, remote working seems recently to spread into more precarious, temporary, and lower-paid jobs, even in outer areas.

Both Vilhelmson and Thulin [34] and De Graaff and Rietveld [119] show that individual propensity to telework is strongly correlated to income and computer access, while other factors seem to play a less important role. The importance of income can be explained by two concurring elements:

- teleworking is more likely to occur in those jobs that require high schooling levels, which typically coincide with higher wage rates;
- high-wage earners are more likely to substitute commuting with leisure time.

The selective nature of these factors explains why remote workers are usually found in high-paid high-tech jobs, more frequent in thick labour markets [37,118,190,221], such as cities or metropolitan areas, while they are less common in rural areas, as it was hypothesised in the first period of development of this concept [222].

A further factor influencing the propensity to telework is commuting distance. Peters et al. [113] show that Dutch teleworkers have longer commutes on average, while Lister and Harnish [223] highlight that in the USA, telework is more extended in the cities with the worst congestion or longest commuter distances. Many other studies [28,102,113,224] find a relation between the home–work distance/travel time and telework practices, which seems to become stronger after a threshold of around 30 minutes [113]. Moreover, home-to-work distance seems to influence not only the likelihood to telecommute but also the telecommuting frequency [225–230].

On these bases, it is possible to argue that remote working has a high level of professional and spatial selectivity: it is more frequent in larger companies, where it is often limited to specific professional categories [28,37,49,102,113,190], and in the case of longer home-to-work travel, which occurs mainly in major cities.

The correlation between the propensity to telework and the dimensions of the metropolitan area is confirmed by several studies. Already, White et al. [181] show that in 2002-04, the proportions of teleworkers for Greater London and the South East were slightly higher than in the other parts of the UK. On the other hand, referring to the French situation, Aguilera et al. [37] find that Parisians are significantly over-represented within the population of French teleworkers: in the very municipality of Paris, their proportion reaches 18%, which is more than double the national average. This result mirrors the concentration of highly skilled professionals combined with bad mobility conditions in the Ile-de-France region (where the share of inhabitants spending over 2 hours travelling per day reaches 22%, against 12% elsewhere in France).

The variety of situations covered by different definitions of telework, as well as the poor measurement of remote working practices, explains the gap between the figures reported by various surveys focused on its diffusion [37,231,232]. However, before the pandemic, these surveys generally returned a low incidence of (full-time) teleworkers, which only rarely exceeded 2 or 3% [119,233].

Empirical evidence of the level of adoption of telework in several European countries has been found by Brewster et al. [234] using the 1995 Cranet-E survey, which was directed to the most senior human resource managers of organisations with more than 200 employees. The largest proportion of teleworkers was reached in Sweden, where nearly half of the employees experienced teleworking, although only 5% for more than 10% of work time. High levels of adoption were also found in other Nordic countries such as the Netherlands and Switzerland (see Figure 2).

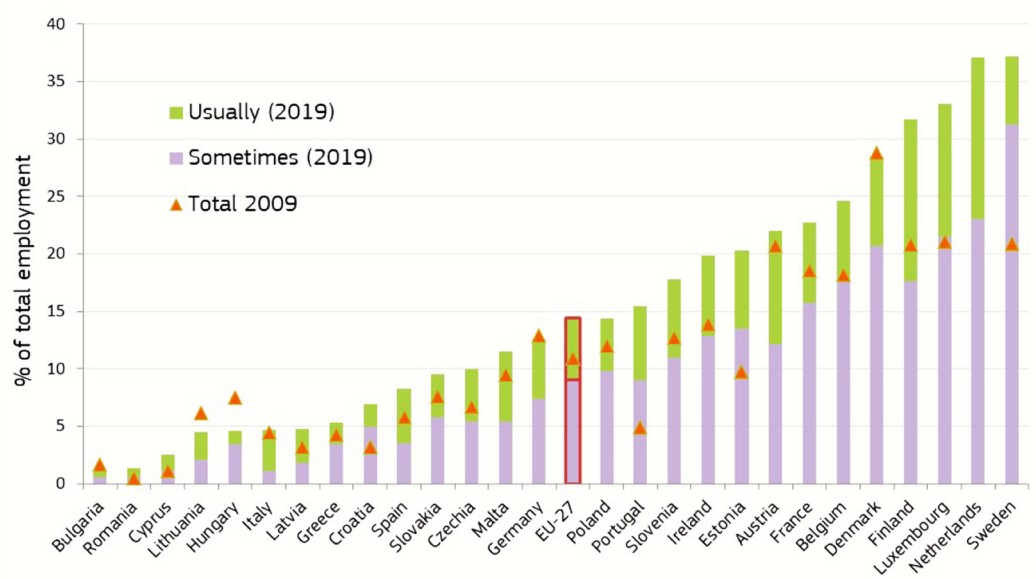

**Figure 2.** Estimated teleworking in and of total labour force. Source: JRC [235].

Another survey, conducted a few years later directly by the European Commission [236], confirmed that telework is more adopted in the Scandinavian countries, the Netherlands, and Switzerland. The United Kingdom and Germany were still above the European average, while France, Spain, and Italy had the lowest proportion. According to this survey, around 4% of the European labour force are regular teleworkers, while 2% occasionally work at home [118]. These differences could probably be explained by both the internal and the external constraints, such as psychosocial factors limiting the propensity to telecommute and the differences in organisational culture among European countries, respectively.

Further surveys and analyses found slightly greater figures; they highlighted that when also considering partial teleworkers (<30% of working time), the incidence in the total workforce tends at least to double (see Table 4).

Nevertheless, considering the few cases in which a comparison of different years is possible, a slight increasing trend appears, especially among regular teleworkers. It can therefore be argued that, before the pandemic crisis, considerable scope existed for further extension of remote working [181], which was nevertheless counterbalanced by several constraints.

**Table 4.** Incidence of regular and total teleworkers in the total workforce in various countries and pre-COVID years.

| Incidence of Telework | | | | |
|---|---|---|---|---|
| **Country** | **Year** | **Regular** | **Total** | **Source** |
| USA | 2002 | | 5% | Todd [237] |
| | 2005 | | 17% | Todd [237] |
| | 2010? | | 24% | Noonan and Glass [174] |
| Canada | 2019 | | 15% | Tremblay [9] |
| Australia | 2006 | | 6% | Shieh and Searle [180] |
| Sweden | 1999 | 4.8% | | Vilhelmson and Thulin [34] |
| UK | 2002 | 3.1% | 7.5% | White et al. [86], Aguilera et al. [37] |
| | 2008–09 | 5% | 11% | White et al. [86] |
| Eire (Rep. of Ireland) | 2019 | | 14% | Redmond and McGuinness [238], Crowley et al. [3] |
| Netherlands | 1995 | 2.3% | | Willigenburg and Van Osch [239] |
| | 2000 | 6% | | Willigenburg and Van Osch [239] |
| | 2002 | | 9% | Todd [237] |
| | 2005 | | 21% | Todd [237] |
| France | 2008 | | 6.8% | Aguilera et al. [37] |
| Switzerland | 2000 | 4.5% | 10% | ECaTT [236] |
| | 2010 | 4.3% | 16.6% | Ravalet et al. [146] |
| | 2015 | 8.8% | 19.3% | Ravalet et al. [146] |
| Spain | 2018 | 4% | 7.4% | Belzunegui-Eraso and Erro-Garcés [1] |
| Italy | 2020 | 3% | 7.4% | De Masi [240] |

The pandemic crisis completely changed these trends, with a sudden increase in the incidence of teleworkers to around 40% in Canada [9] and to similar figures in Italy, where the number of remote workers passed from 570,000 to 8 million [240].

*3.2. Direct Effects of Telecommuting*

One of the most frequently invoked benefits of remote working is the reduction of road traffic and its correlated environmental impacts. The idea that working from home can be a way to cope with road congestion and/or energy consumption, greenhouse gas emissions, and air pollution generated by physical commuting has been debated since the 1990s [77,241–243]. It is well-represented in many urban transport plans or air quality programs throughout the world.

This viewpoint finds many empirical confirmations. A number of different researches have focused on the potential of telecommuting to reduce vehicle kilometres and their impacts in order to reduce energy consumption and air pollution [27,78,81,109,181,243–256], as well as travel time, which is a social benefit in itself. US studies state car mileage decreases between 48% and 77% on teleworking days, with a weekly decrease between 9% and 11% when traditional working days are included [27,244,245]. The SUSTEL study found an average reduction in commuting distance of 98 km/week for telecommuters. In Switzerland, Ravalet et al. [146] find that travelled distances in telecommuting days (54 km) are shorter than in normal working days (66 km); this also implies a reduction in travel times (from 119 to 98 minutes). Lachapelle et al. [77] found that, in Canada, working only from home for one day/week does reduce overall travel by 13 minutes. A recent study on remote working in Sweden finds that telecommuting trips are fewer and shorter, and that teleworkers use active transport modes (biking or walking) more often than non-telecommuters with lower car dependence [257].

The total amount of mileage reduction due to telecommuting also depends on the average travelled distance, which tends to be greater for teleworkers than non-teleworkers. Ravalet et al. [146] show that in Switzerland, the house–workplace distance is 32.3 km for teleworkers and 25.4 km for non-teleworkers. Another study of the same authors, developed on the basis of the Swiss Mobility and Transport Microcensus, finds that telecommuting involves longer commute lengths than normal jobs and also that this difference is increasing over time [258]. Quite similarly, Van Ham et al. [259] highlight that Dutch teleworkers increase their commute length by 12 percent on average. All conditions being equal, the difference in travelled distance means that the mileage reduction generated by telecommuting is more than proportional to the variation in the number of home–work travels.

Moreover, scholars have often underlined that telecommuting tends to act mainly on rush hour, with amplified benefits: fewer peak hour commute trips generally result in a reduction in congestion, smoother flows, and fewer emissions for a travelled kilometre [81,181]. Hamer et al. [109] highlight that telecommuters reduce the number of daily trips by 17% and the distance travelled by 16% as an average, and also that in peak hours, these variations are greater (19% and 26% respectively), with a maximum of −34% for car users. Studies in the US show that full-day remote workers travel less on telecommuting days, while part-day remote workers only decrease the number of trips during rush hours [260].

Nevertheless, it is clear that the potential energy and emissions savings through telecommuting greatly depends on the commute modal split [219]. White et al. [181] point out that in the case of public transport, resource savings are only made when a change to the pattern of timetabled service occurs, at least in terms of reduction of peak capacity (such as train length). In this case, telecommuting can result in more of a change of the public transport market than in environmental benefits. Nevertheless, indirect social benefits can appear, such as reduced crowding on existing services and/or the possibility to meet currently frustrated demand.

The impact paths of telecommuting are even more complex in the case of telecentres. Already, Mokhtarian [176] shows that working from a local or a neighbourhood work centre can reduce physical commuting while avoiding the faults of home working (e.g., isolation, lack of focus). Anyway, the estimate of the potential for reducing kilometres became more complex, partly because of possible changes in mode choice [31,244,261,262]. Studying the case of Stockholm, Bieser et al. [263] highlight that average travel time is significantly shorter on days when commuter worked from a telecentre instead of the main office, but they find no evidence that working from the telecentre induces a major shift to more energy-intensive transport modes.

### 3.3. Rebound Effects

The analysis of the rebound effect attributable to the spread and consolidation of telework quotas is obviously connected to the dynamics and elements of evaluation outlined above, which consider the changes induced on the behaviour and habits of mobility and their consequences in terms of energy consumption and environmental externalities. The considerations developed here draw on the studies and analyses carried out in the various international contexts of experimentation concerning telework and direct the considerations about the territorial effects determined by the structural introduction of telework to the Italian national context. It should be noted that the characterisation of the settlement structure is very regionalised, with metropolitan areas intermingled with internal areas.

A second type of rebound effect relates to the loosening that telework determines of the constraint of residency. The COVID-19 pandemic crisis has imprinted a modification of the system of preferences, both in terms of the configuration of the spaces of one's own home (seeking adequate spaces for work and breathing space, such as gardens and balconies) and in terms of a tendency to reconsider housing options that are less close to urban/metropolitan centres but with a better offer of landscape—environmental potential effects. It should be considered that a typical repositioning of one's own home draws

wider functional geographies and tends to be more articulated: from the metropolitan area, one turns to more pleasant contexts, with a greater presence of greenery and parks, and therefore tends to be in low-density areas, a condition of urban diffusion that determines wider travel radii for access to the system of services.

### 3.3.1. Rebound Effect: Behavioural Changes and Effects on Mobility Demand

The first important rebound effect to highlight is the change in behaviours of teleworkers: teleworking reduces spatial and temporal constraints on individuals' willingness to engage in activities in different locations within a given time frame.

Although increased teleworking can generate direct and immediate environmental benefits, as evidenced by the COVID-19 crisis [264] (74–77), in the long run, the "rebound effects" associated with increased non-work-related travel, farther residential relocation, car dependence, and different consumption patterns for teleworkers can reverse all benefits [221,265–267].

Many researchers have found that reducing the frequency of commuting movements does not necessarily imply a reduction in miles travelled [217,224,267,268]. At the same time, several studies document mobility behaviour based on more or less constant temporal availability [269,270]: individuals thus appear to be oriented toward establishing a "travel time budget" [271].

Another potential rebound effect of telecommuting is that it may stimulate more non-work trips. In this case, the reduced demand for mobility due to reduced daily commuting is partially or fully offset by additional travel for other purposes. This is sometimes referred to as a "complementary" effect of telework [272,273]. More recently, Silva and Melo [90] show that UK teleworkers reduce the number of commuting trips but not the overall weekly travelled distances, and that they tend to increase miles travelled by car. This result seems to be confirmed by Ravalet and Rérat [258], whose study highlights that, in the Swiss case, non-work travels on telecommuting days partially compensate for the absence of commute movements, resulting in distances travelled per week longer than those of non-telecommuters.

The adoption of telecommuting correlates with less proximity between workplaces and residence, which risks generating spatial dispersion with more car trips for other reasons, creating increased car dependence. A fairly clear link has been established between increased online access to work and reduced proximity between places of residence and work; teleworkers increase their search space for a residence and move farther away, causing greater commuting distance on non-telework days [111,118,274], and increased distance may at least partially offset the reduction in the number of commuting trips.

However, it seems clear that in some cases, the increased adoption of part-time teleworking could increase weekly, monthly, or annual commuting travel. More generally, the environmental benefits of telecommuting will depend not only on the frequency of telecommuting, but also on the distance travelled by telecommuters from home to their place of work [77].

Moreover, evidence from both the United States and Europe highlights that teleworking can induce long-term changes in residential location, and that this effect can offset some of the environmental benefits [225–227,267,275]. The size of this rebound effect may change depending on local factors, such as the r differential between urban and peri-urban regions in real estate prices and the generalized cost of commuting. Nevertheless, Collantes and Mokhtarian [228] underline that the possibility to telecommute contributes to the decision to relocate only for a small share of teleworkers, but also that those workers relocated significantly farther away from their workplace. Muhammad et al. [123] show that, in the Netherlands, telecommuting has enabled people to commute longer distances, and Hergheth [230] found that in the German case, telecommuting after a move or a job change highly increases travel distance and time.

Greater spatial dispersion of workers, when traced to housing preferences for larger dwellings in greener settings, results in a spread to more suburban areas that are likely to

be more car-dependent [90,218,276], short of areas located along regional transportation power lines (particularly rail lines).

According to Tremblay [9], the coming years will certainly see an increase in telecommuting, which could lead to a relocation to the suburbs and small towns of the regions. We will then have to consider the trade-off between the positive effects of devolution (reduced pollution from cars and reduced need for infrastructures such as roads and public transport) and the negative effects (increased need for digital infrastructure and increased greenhouse gas emissions associated with so-called "digital pollution"). Spatial reorganisation related to the increase in teleworkers could then be structured around telework centres [9]. Once the measures related to the COVID-19 epidemic have been overcome, it is indeed conceivable that people who choose to move to areas more distant from their employer's offices will tend to prefer a "neighborhood" coworking space in order to avoid working in isolation at home and to take advantage of the professional networks, computer facilities, meeting rooms, or other facilities that coworking spaces can offer.

A perspective on possible scenarios generated by the COVID-19 emergency is suggested by Spadaro and Pirlone [277]. These authors outline four possible scenarios: (i) Return of the urban mobility system to the pre-coronavirus-disease-2019 (COVID) situation; (ii) Prevalence of demand for private mobility (use of the car); (iii) Reduction in the demand for mobility; (iv) Achievement of integrated multimodal mobility. The four scenarios consider the different options between return to pre-pandemic conditions, increase or decrease in demand for mobility, and possible modal split of transport (related to fragility and fears matured for collective transport).

### 3.3.2. Rebound Effect: The Transformation of Inhabitants' Settlement Preferences and Spatial Effects

Cities are facing major structural challenges, as many are wondering if they will still be as attractive as before, with telework being more widely used and rural areas proving more resilient in cases of similar crises. The redesigning of public spaces, the "15-minutes city" concept, and better-coordinated governance seem to be necessary responses in order to maintain cities' competitiveness and attractiveness [278] (p. 36).

The COVID-19 pandemic crisis and the resulting socio-spatial practices trigger forced rethinking of spatial planning approaches. The serious risk of the coronavirus, no doubt, challenges at least some existing conventions and ignites the rethinking of future urban forms [279]. This process presents an opportunity for planners and urbanists to question some of their fundamental social and spatial assumptions, thus participating in the development of a new socio-spatial order.

Certainly, the debate around the changes triggered by the pandemic crisis brings to mind the search for historical balances and reference models, such as the Garden City proposed by Ebenezer Howard, aimed at achieving "a healthy, natural, and economical combination of city and country life". Sturzaker [280] reminds us that in the British case, there is very little "country life" in the converted office blocks and newly built apartments that are being constructed in British cities. On the other hand, it is worth remembering, again with reference to the fragility of territorial organisation, how research has documented how contagion has been more relevant precisely in low-density peri-urban realities [281,282].

Confidence in the possibility of rethinking the form and organisation of cities finds conflicting opinions; some argue that there has never been an effect of a pandemic that caused rethinking the forms of the city in history nor now a macro-change in metropolitan geographies, limiting the estimate of the effects of the pandemic crisis to a simple, albeit important, acceleration. Within the same reflection, however, Florida recognises remote work as the main "pull force" in the transformation of cities in the coming years, both in settlement geographies, with the possibility of choosing to move away from urban centres for a portion of the population, and in the potential trade-off between the demand for housing more suitable to host work activities and the reduced need for office space

for companies; however, on the other hand, it sustains the presence of "push factors", in particular those linked to the preferences of the young population for urban realities full of life, relationships, and opportunities.

## 4. Teleworking Potential and Its Effects in Italy: A First-Level Assessment

Scientific literature provides a wide set of elements related to the propensity for teleworking and its potential in terms or reduced mileages and air pollution impacts, as well as the possible rebound effects reducing this potential in the medium or long term.

This study aims to develop a first-level assessment of these main results in the case of Italy, which has been probably not so interested in telecommuting in the past, with quite poor interest for the theme and therefore a lack of reliable results about its diffusion and potential.

The rationale of the assessment is based on the three logical levels presented above, namely (A) the telework potential in different parts of the country, (B) its effect in terms of mileage and pollution, and (C) the possible indirect effects leading to structural change in the urban settlements.

More specifically, this first-level assessment of possible effects of teleworking in Italy aims to verify three basic hypothesis:

(A) telework potential tends to be higher in metropolitan areas, where white-collar employment is more oriented to new technologies;

(B) therefore, direct effects on commuter mobility are mainly located in catchment areas of bigger cities;

(C) on the other hand, rebound effects are linked also to the distribution of holiday homes, which is highly selective throughout the country; this condition alternatively strengthens or weakens secondary effects at regional level around main cities. The methodology and the results of the study are explained below.

### 4.1. Teleworking Potential

The first goal of this study is to begin the development of a statistically reliable framework of teleworking (TW) propensity in the whole of Italy. This can be done by assigning a specific level of TW propensity to each economic sector (following ATECO/NACE classification): "very high" for technical, ICT, and publishing activities, "high" for other technical services, "medium" for industrial activities, and "low" for other services not involving individual/personal care or direct interactions, such as education, health, or tourism (see Table 5).

This classification allows for a first estimate of employments belonging to activities with different levels of TW propensity, on the basis of the last census (2011) and its update by ASIA archives (see Table 6):

- very high propensity is limited to nearly 3% of the total private workforce (half a million people), with an increasing trend from 2011 to 2017;
- high propensity involves 14% of the total private workforce (nearly 2,5 million people) in a context of stability;
- medium propensity incidence is reducing from 25% to 23% (around 4 million people), mainly because of industrial crisis and restructuring, which is lowering employments in many parts of the country;
- low propensity appears to be a little bit more marginal (12% of total workforce, more or less 2 million people) and stable.

**Table 5.** Telework propensity by economic activity.

| Class | TW Propensity | Economic Activities (NACE rev.2) |
|---|---|---|
| 4 | Very high | 58 Publishing activities; 62 Computer programming, consultancy, and related activities; 63 Information service activities; 73.2 Market research and public opinion polling; 74.3 Translation and interpretation activities; 82.2 Activities of call centres |
| 3 | High | 41.1 Development of building projects; 46.1 Wholesale on a fee or contract basis; 59.1 Motion picture, video, and television programme activities; 59.2 Sound recording and music publishing activities; 60 Programming and broadcasting activities; 61 Telecommunications; 64 Financial service activities, except insurance and pension funding; 65 Insurance, reinsurance, and pension funding; 66 Activities auxiliary to financial services and insurance activities; 69 Legal and accounting services; 70 Activities of head offices, management consultancy activities; 71.1 Architectural and engineering activities and related technical consultancy; 73.1 Advertising; 74.1 Specialized design activities; 74.2 Photographic activities; 74.9 Other professional, scientific, and technical activities n.e.c.; 82.1 Office administrative and support activities; 82.3 Organisation of conventions and trade shows; 82.9 Business support service activities n.e.c.; 84 Public administration and defence, compulsory social security; 85.5 Other education |
| 2 | Medium | All manufacturing activities (from 05 to 39) |
| 1 | Low | 01.6 Support activities to agriculture; 02.4 Support services to forestry; 45.1 Sale of motor vehicles; 45.3 Sale of motor vehicle parts and accessories; 45.4 Sale, maintenance, and repair of motorcycles and related parts and accessories; 46.2 Wholesale of agricultural raw materials and live animals; 46.3 Wholesale of food, beverages, and tobacco; 46.4 Wholesale of household goods; 46.5 Wholesale of information and communication equipment; 46.6 Wholesale of other machinery, equipment, and supplies; 46.7 Other specialised wholesale; 46.9 Non-specialised wholesale trade; 47.4 Retail sale of information and communication equipment in specialised stores; 47.6 Retail sale of cultural and recreation goods in specialised stores; 68 Real estate activities; 71.2 Technical testing and analysis; 72 Scientific research and development; 77 Rental and leasing activities; 78 Employment activities; 79 Travel agency, tour operator, and other reservation service and related activities; 80.2 Security systems service activities; 80.3 Investigation activities; 81.1 Combined facilities support activities; 85.4 Higher education; 90.0 Creative, arts, and entertainment activities; 92.0 Gambling and betting activities; 93.2 Amusement and recreation activities |

Elaboration on ISTAT data.

**Table 6.** Employments by telework propensity—Whole of Italy (private sector).

| TW Propensity | 2011 | | 2017 | | Variation 2011–17 | |
| :---: | :---: | :---: | :---: | :---: | :---: | :---: |
| | *Employments* | % | *Employments* | % | *Employments* | % |
| very high | 456,234 | 2.8% | 512,166 | 3.0% | +55,932 | +0.2% |
| high | 2,371,428 | 14.4% | 2,451,490 | 14.4% | +80,062 | −0.1% |
| medium | 4,171,903 | 25.4% | 3,981,929 | 23.3% | -189,974 | −2.1% |
| low | 1,986,209 | 12.1% | 2,170,010 | 12.7% | +183,801 | +0.6% |
| **Total** | **8,985,774** | **54.7%** | **9,115,595** | **53.4%** | **+129,821** | **−1.3%** |
| no propensity | 7,435,766 | 45.3% | 7,943,819 | 46.6% | +508,053 | +1.3% |
| **Total workforce** | **16,421,540** | **100.0%** | **17,059,414** | **100.0%** | **+637,874** | **+0.0%** |

Elaboration on ISTAT data.

However, it is worthwhile to point out that economic sectors with no propensity to telework employ a growing workforce, mainly as a result of the increase in personal care jobs. The consequence is that no particular "drag effect" seems to appear in the Italian job market fostering telecommuting.

A first-approximation assessment of TW potential can subsequently be obtained comparing TW propensity with origin–destination (OD) matrix of Home-to-Work movements between all Italian municipalities.

A basic hypothesis is that TW propensity is a factor influencing the frequency of remote working (from an average 1 day/week for low propensity to 4 day/week for very high propensity), and that the choice to actually telecommute is linked to travel time, too. Therefore, it is possible to make a comparison between telecommuting base frequency and travel times classes used in the Census OD matrix (0–15 min, 15–30 min, 30–60 min, and more than 60 min, referring only to outbound movements), in order to estimate time savings generated by telecommuting.

Taking 4 h/week of time-saving as a functional threshold, it is possible to argue that people working in sectors with very high TW propensity could decide to telework with high frequency (4 day/week) with relatively low travel times (15–30 min), whereas people working in sectors with medium TW propensity tend to telecommute less frequently (2 day/week), only in the presence of long travel times (>60 min). These quite conservative parameters return a TW potential of nearly 1.16 million outbound movements for each working day throughout Italy.

Figure 3 illustrates this potential by destination (i.e., the municipality of the workplace before telecommuting): this result tends to confirm the tendency to concentrate TW potential on major cities (such as Milan, Rome, and Turin), which has already been highlighted in other studies for the UK [181] and France [37].

Nevertheless, some TW potential seems to also exist in several rural areas of some regions (Emilia-Romagna, Toscana, Puglia).

Anyway, it is worthwhile to underline that this preliminary result has only indicative value, since it does not take into account other socio-economical indicators (age, gender, type of employment), which can obviously play an important role in the decision to telecommute or not.

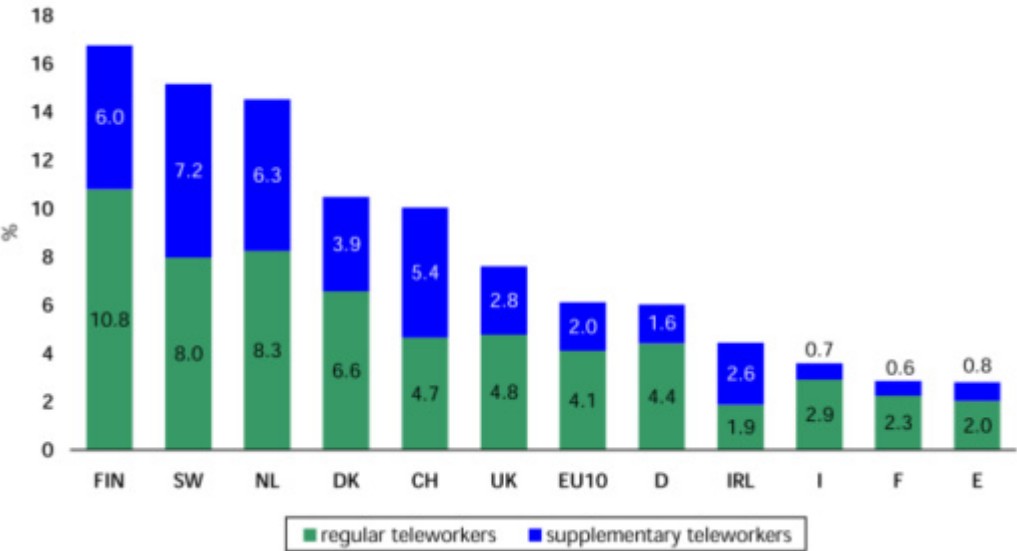

**Figure 3.** Regular and supplementary teleworkers in % of total labour force. Source: ECaTT [236].

### 4.2. Direct Impact of Telecommuting on Mobility Demand

In the Italian case, the first approximation estimate of the likelihood to telecommute enables mapping the distribution of teleworkers' places of residence (Figure 4). These places tend to aggregate mainly in outer parts of metropolitan areas, i.e., in the municipalities that are simultaneously:

- close enough to the main aggregations of workplaces with high teleworking potential;
- far enough from these workplaces to encourage telecommuting.

At the same time, highly rural areas tend to be negatively affected by the distance from high potential workplaces, which reduce the likelihood of normal commuting on these desired lines.

This result is consistent with some other researches, which underlines the likelihood of telecommuting reaching its maximum for people living in metropolitan or regiopolitan areas [230].

One important consequence is that travel reduction tends to especially affect car or public transport journeys between outer metropolitan areas and city cores. This can result in emphasising the impact of teleworking on road congestion and/or public transport crowding (see Figure 5).

### 4.3. Rebound Effect on Settlement Patterns: A Territorial Scenario for the Italian Context

Declining the general considerations developed above, open to contributions from very different contexts, the reflections on the rebound effect dropped into the context of the Italian territory cannot but assume some specificities that also represent important opportunities.

The Italian reality records some distinctive elements of interest: (i) a complex settlement structure of the metropolitan systems, with an articulation of the "internal areas" that makes them accessible in reasonably considerable times in the geographies of telework that see as magnets the main urban contexts; (ii) a substantial building heritage consisting of disused or obsolete buildings, but also of many tourist homes or "second homes" in coastal, mountain, or lake contexts, which could partly change status; (iii) a specific policy (National Strategy for Inland Areas) aimed at the development of inland areas, which have long presented problems of economic weakening and depopulation, which could take advantage of current trends to support a strengthening of infrastructure and services that represent the prerequisite for a demographic and economic recovery.

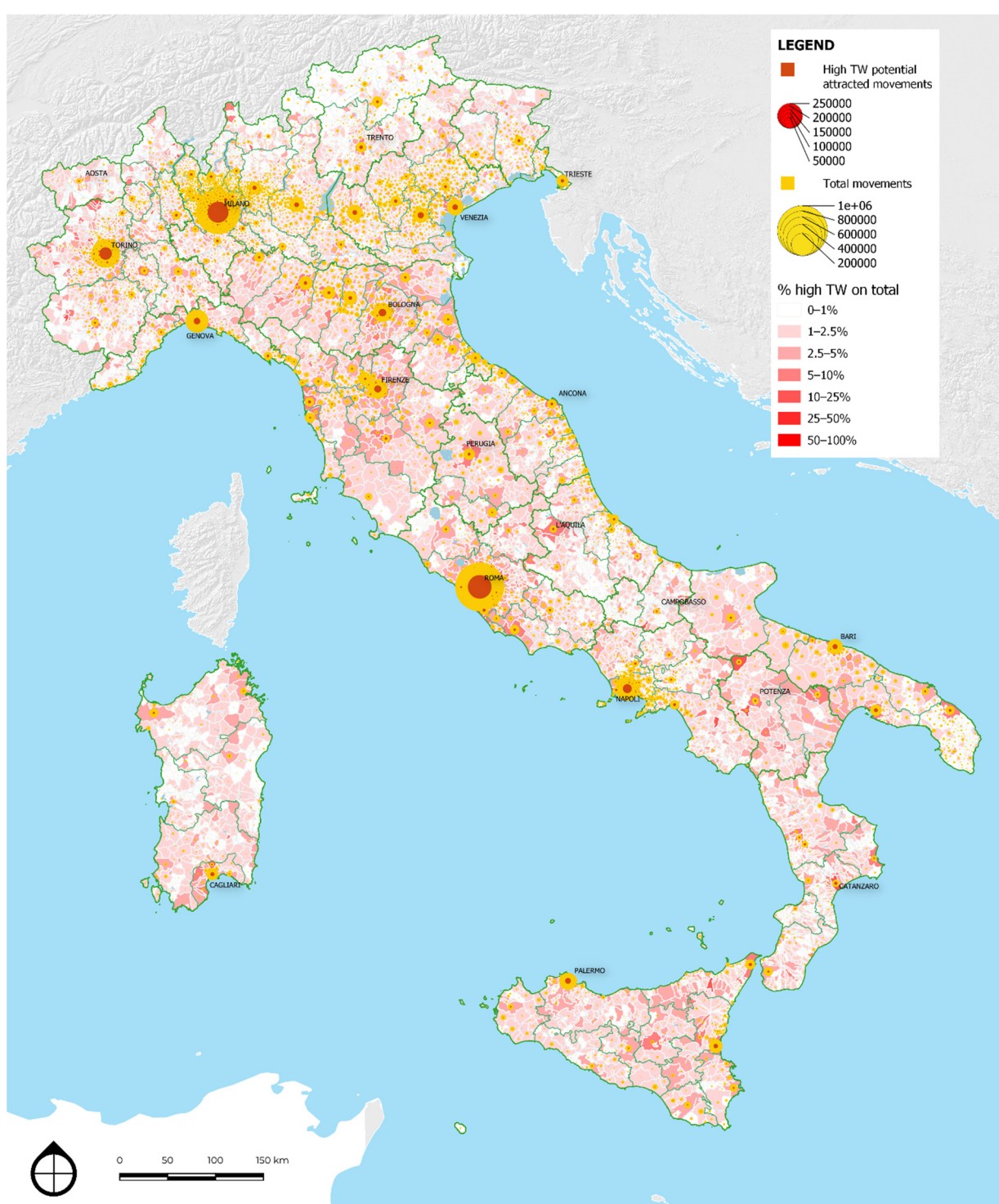

**Figure 4.** Teleworking potential by workplace. Elaboration on ISTAT data.

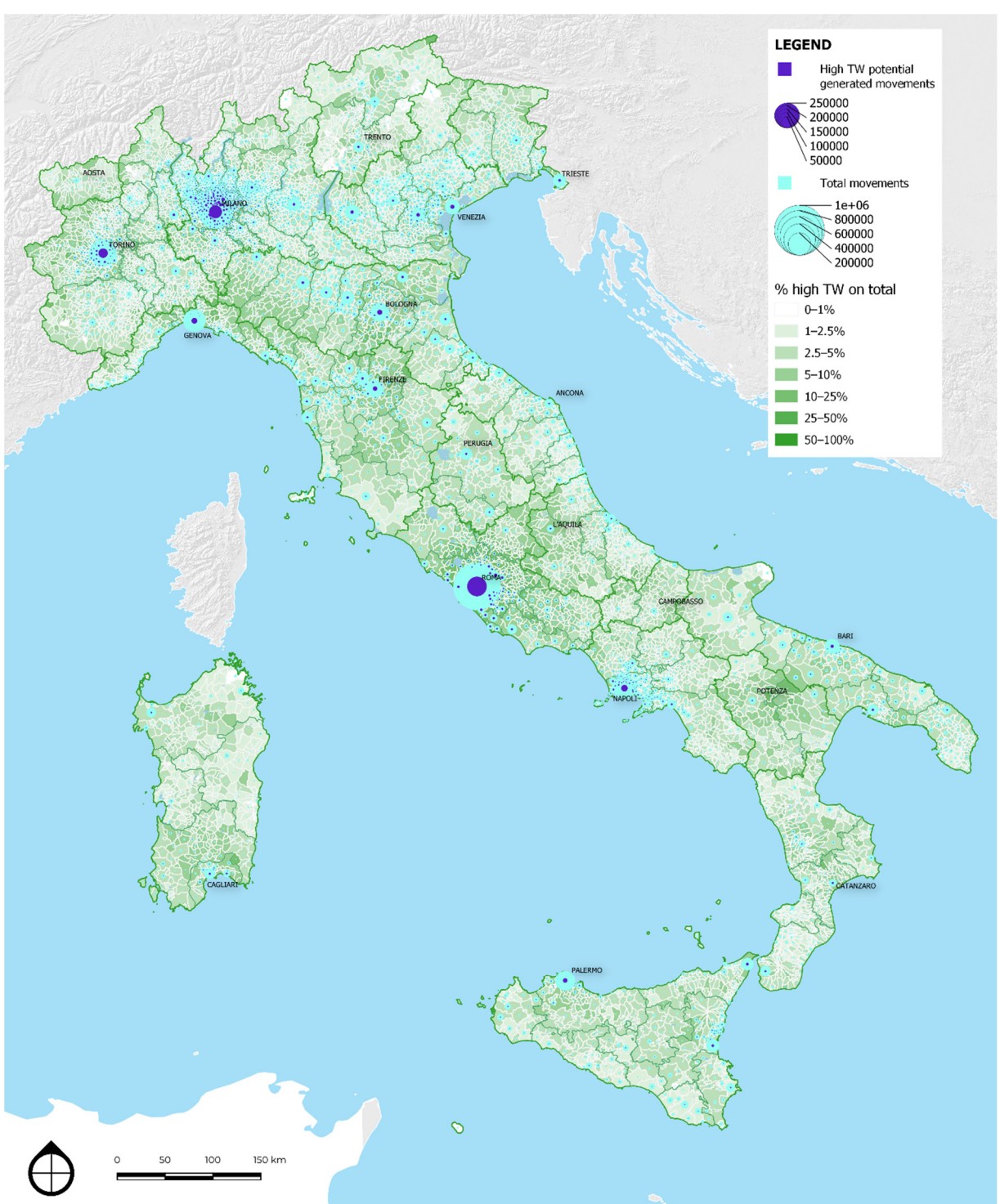

**Figure 5.** Teleworking potential by place of residence. Elaboration on ISTAT data.

From this viewpoint, it is possible to hypothesise that telecommuting will foster a gradual relocation of residences from outer metropolitan areas further away. In the Italian case, this could mean a residential move towards the closer part of internal areas, such as the lower alpine valleys, which often lie between 50 and 100 km from major cities of Northern regions, such as Milan, Turin, Bergamo, Brescia, and Verona.

During recent decades, at the time of industrial growth after the Second World War, these areas often suffered a depopulation process, leaving an important housing stock,

gradually transformed into holiday dwellings, generally owned by families that lived in the metropolitan areas (see Figure 6).

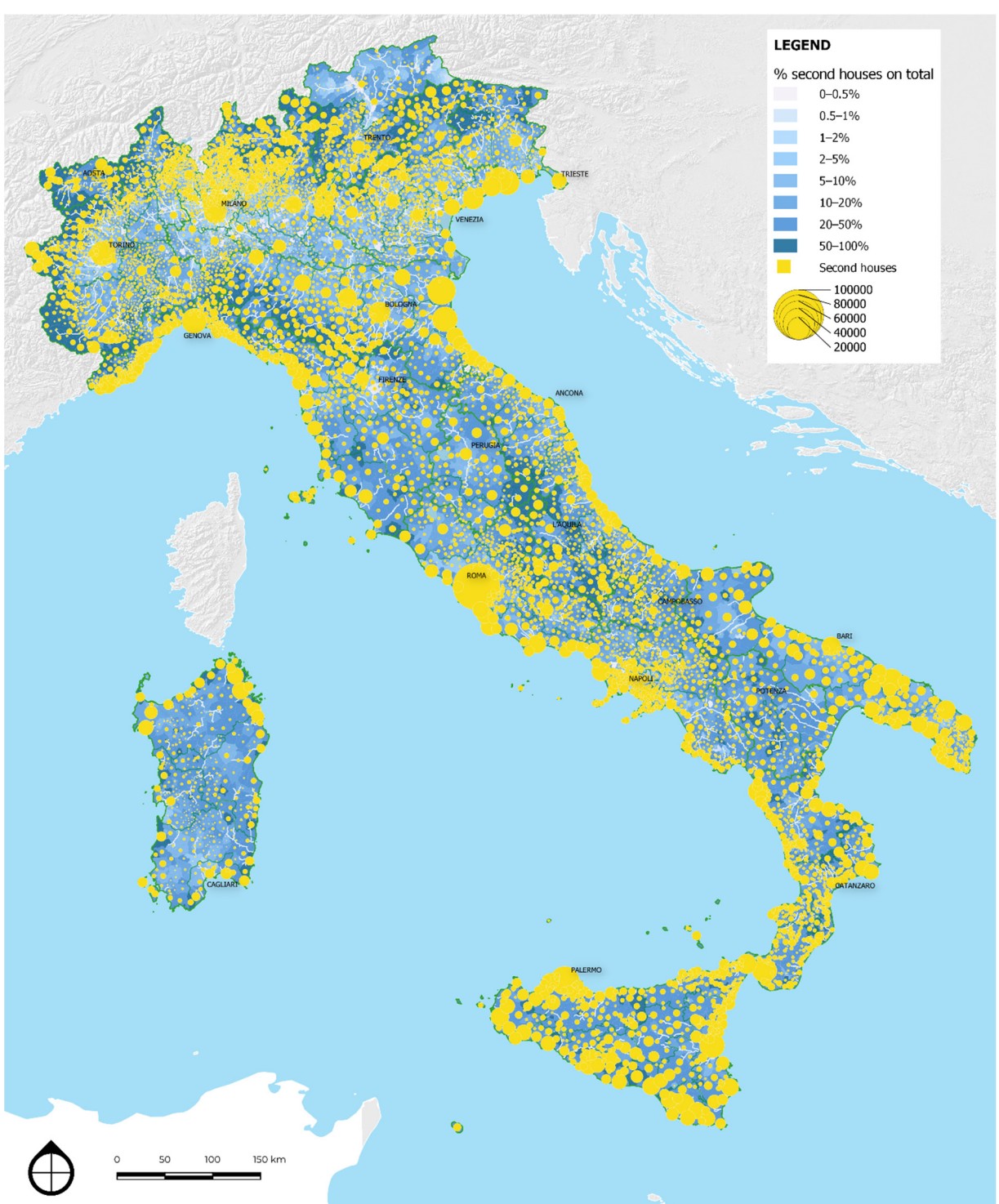

**Figure 6.** Second homes. Elaboration on ISTAT data.

A possible research program deals with the complex functional linkages between high-telecommuting-potential workplaces, current residence places in the outer metropolitan areas, and holiday house stocks in the lower alpine valleys (or other internal areas in central/southern Italy). A careful comparison of the distances, costs, and conditions of (tele)commuting from metropolitan or mountain residences can highlight the likelihood

of a possible strengthening of current marginal areas, assessing at the same time new commuting practices and configuration in non-teleworking days.

A detailed comparison among high-TW-potential workplaces and places of residences in metropolitan areas, as well as second houses in surrounding regions, can be the basis for increased possible relocations of people and changes in (tele)commuting structures.

In the case of Milano (see Figure 7), clear evidence arises of a great density of high-TW-potential workplaces in the very city centre and a few peripherical poles, faced by a wide number of metropolitan municipalities with medium TW potential for their residents. On the other hand, many local districts in the lower Alps and around Como Lake have a wide range of second houses, which could quite easily be chosen as alternative places of residence by teleworkers.

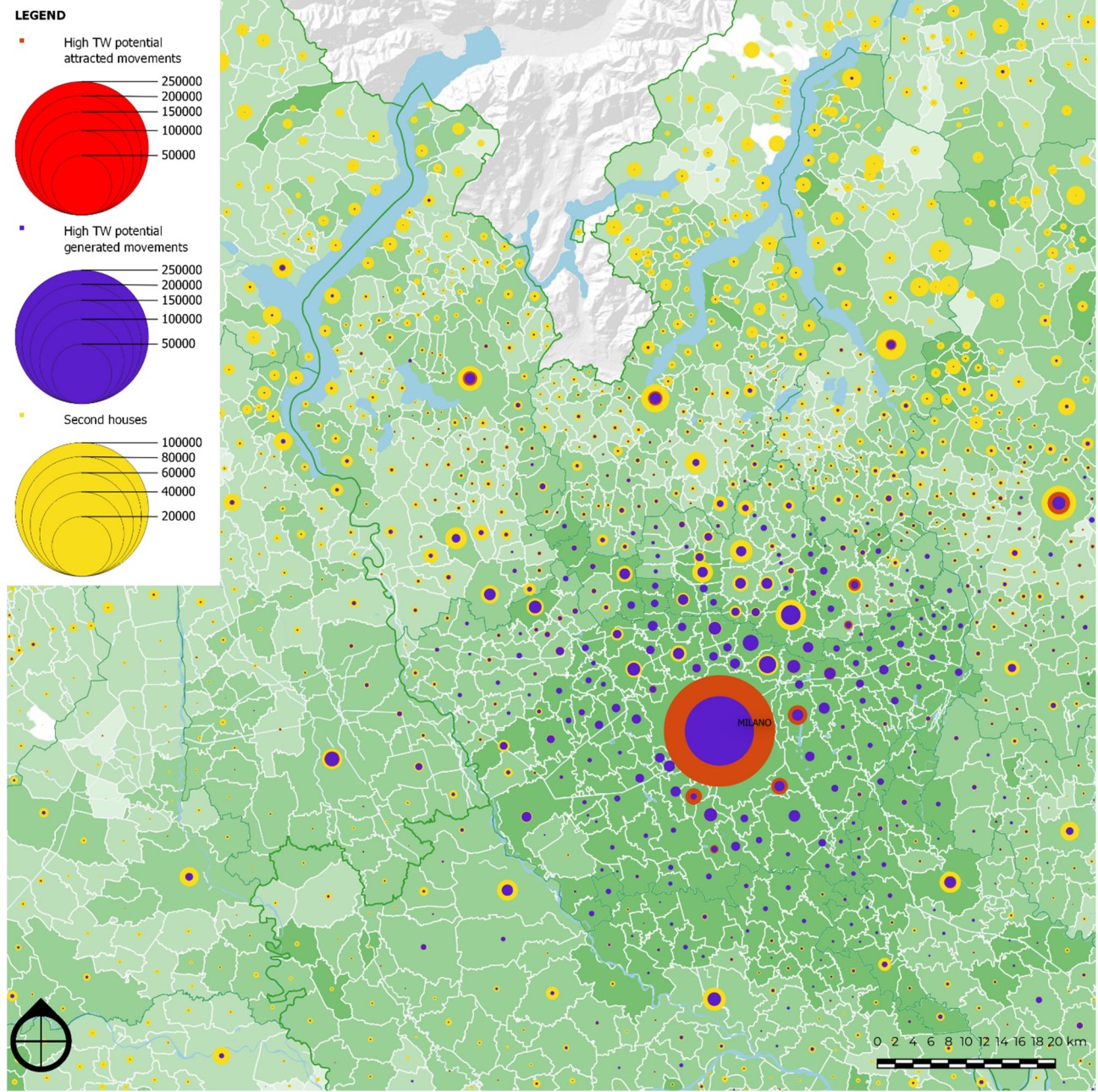

**Figure 7.** High-telecommuting-potential workplaces, places of residence, and second houses: Milano. Elaboration on ISTAT data.

A quite similar situation arises in the case of Torino (see Figure 8), with a good concentration of high-TW-potential employments in the city, a metropolitan belt marked by high-TW-potential places of residence, and a great amount of second house throughout the Alps to the French border.

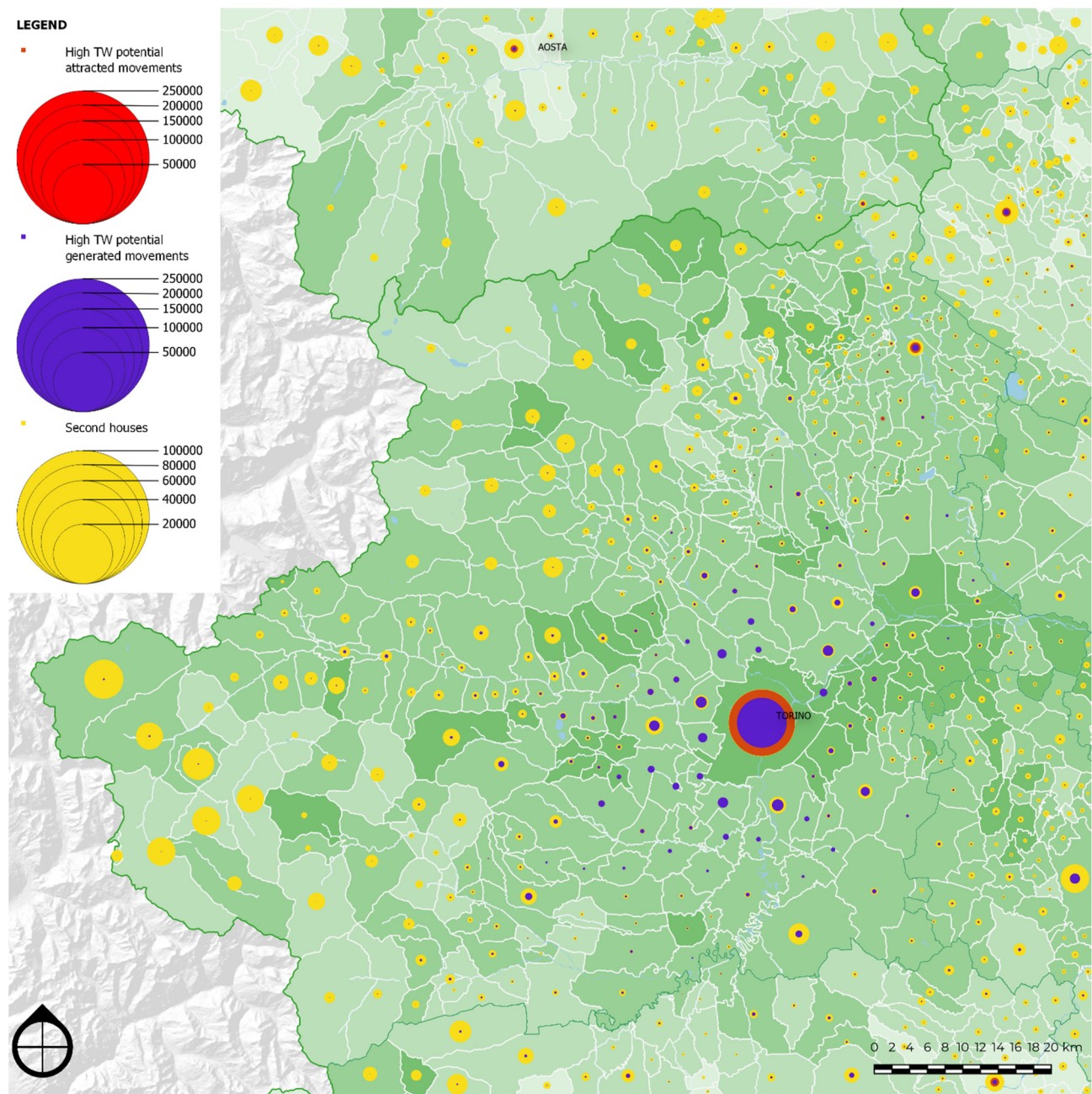

**Figure 8.** High-telecommuting-potential workplaces, places of residence, and second houses: Torino. Elaboration on ISTAT data.

Another interesting case is Brescia (see Figure 9), whose smaller, but not negligeable, amount of high-TW-potential employment is met with a great amount of second houses around the Garda and Iseo lakes.

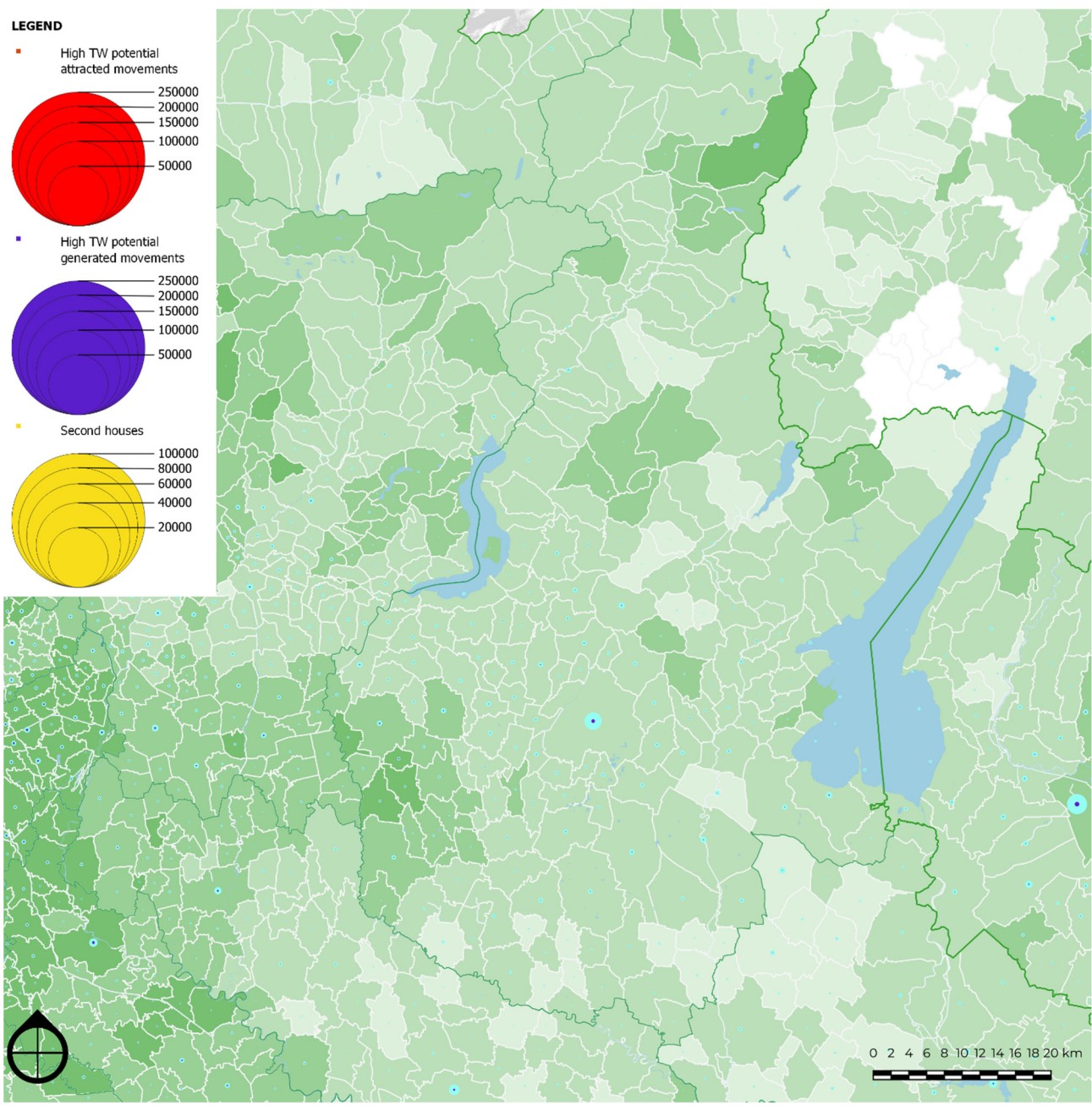

**Figure 9.** High-telecommuting-potential workplaces, places of residence, and second houses: Brescia. Elaboration on ISTAT data.

In all these cases, the relocation of telecommuters towards remote places could have the effect of increasing the commuting length in non-telecommuting days. The final outcome will probably be a new type of commuting, less frequent but more oriented to medium or even long range, which would require some important changes in public transport supply (number of stops, commercial speeds, fare structure, etc.).

## 5. Conclusions

Telework and the evolution of the living model: will the health emergency and a forced application of telework through legal provisions be enough to put the issue in structural terms? After decades of rather limited adoption (with a few exceptions), will a scenario of significant application of telework open up?

The option of structural application of telework, to be modulated according to the different specificities and organisational realities of companies (and the preferences of the employees), has among its possible benefits: (i) a reduction of the direct environmental impact related to the transport of people; (ii) an effect on the economic system pushing to evolution of business organisation protocols (from teleworking to smart working); (iii) an effect on the well-being of people, with better ease in combining work time and family life commitments; (iv) a reduction of the spatial constraints related to the workplace and a consequent greater freedom of choice of localisation of one's main home. However, it is unlikely that all these benefits will appear together, because they tend to partially offset each other.

The objective of this work is twofold: on the one hand, to constitute a platform for the extensive literature on the subject, representing both theoretical reflections and evidence of experiences and experiments conducted in different countries; on the other hand, to outline a path of empirical analysis in the Italian national context, aimed at bringing out possible trajectories of evolution of settlement systems—to connect the dense metropolitan regions with the inland areas of reference—in relation to the mobility networks that constitute their connecting framework.

In addition to the theoretical framing of teleworking, this work tried to verify some basic hypotheses about its potential at the national level in Italy through first-approximation statistical data analysis. Thus, it highlights that high-teleworking-potential employments in Italy (1,16 million people) tend to concentrate in biggest cities, without a clear growing trend at a structural level, due to the increasing incidence of personal care jobs. This result is strictly linked to the distribution of direct effects, which seems to be more a matter of metropolitan context than rural area, in accordance with the outcomes of researches developed in other countries. In this situation, rural area have the chance to be affected by secondary or rebound effects only under some conditions, specifically such as the existence of holiday houses stock and a medium level of accessibility at the regional scale.

Considered as a whole, this chain of possible effects underlines the selectivity of smart working as a tool to revitalize rural areas, which can hope to take advantage of telecommuting only under certain, well-defined conditions.

Within this reflection opens the opportunity to propose scenarios for the evolution of settlement geographies based on a lower component of forced mobility and greater freedom in the choice of place of residence. In this sense, it seems interesting to relate the potential demand, generated by the development of telework, with the possibility of settling in contexts of landscape and environmental value, thus investing in areas marginalised by metropolitan development, which often have abandoned or underutilised residential stock (semi-abandoned villages, tourist homes).

The paper makes an analysis of the potential impact of the affirmation of telework in the Italian national context, trying to territorialise the demand for telework and the dislocation of the workforce; this analysis has focused on metropolitan areas that have economic sectors more suited to telework and a commuting area expanded to the metropolitan scale. From the cartographic elaborations, a picture emerges of the significant potential application of telework that concerns the main metropolitan polarities, where functions and services are centralised and present significant quotas of working time that can be operated remotely. Such a situation therefore may increase residence—in a differentiated way—in moderately mountainous or lake areas that possess a building stock of little-used or abandoned tourist residences.

This evolution of the settlement geographies, however, refers to possible further reflections aimed at considering the overall modification of metropolitan systems and the effects produced both on the central areas of business districts (conceived for a high intensity of use) and on some monofunctional residential contexts or poorly qualified suburbs exposed to the competition of more attractive areas in terms of landscape. Still, a dilution of the settlement geographies poses the opportunity to endow the minor urban

contexts with services to persons and, for work, with telework centres at the service of weakly infrastructured places.

Furthermore, the scenario under investigation implies a reorganisation of transport networks and of public transport in particular, which could be called upon to support a lesser share of the daily demand for short-range transport and instead record a less frequent demand with medium-range movements.

In conclusion, this research outlines working trajectories that need empirical analysis in different contexts in order to recognise ongoing trends and to design policies to guide settlement trends that pose, in terms of sustainability and environmental quality, some critical concerns but at the same time contain significant opportunities.

**Author Contributions:** Supervision, project administration, funding acquisition, F.A.; software, investigation, data curation, visualization, A.D.; Conceptualization, methodology, validation, formal analysis, resources, writing—original draft preparation, writing—review and editing, F.A. and A.D. All authors have read and agreed to the published version of the manuscript.

**Funding:** This research received no external funding.

**Institutional Review Board Statement:** Not applicable.

**Informed Consent Statement:** Not applicable.

**Data Availability Statement:** Not applicable.

**Conflicts of Interest:** The authors declare no conflict of interest.

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
