# Peer review of "The Breath of the Metropolis: Smart Working and New Urban Geographies"

_sustainability, doi:10.3390/su14021028_

Round 1

Reviewer 1 Report

The subject is current, relevant, and suitable for the journal. The bibliography considered is recent and adequate. All quotes have sources.

Suggestions for improvement are:

  1. The title of the article does not fully reflect the content of the paper;
  2. After the abstract the keywords should appear;
  3. Table 3.2. does not show frequency values;
  4. Improve the image quality of figure 3.1.;
  5. Lines "459" to "468" are missing the dashes;
  6. It would be important to have a methodology chapter, to bridge the gap between literature review and empirical study;
  7. At the end, the authors should compare (discuss the results)  the study on the potential implementation of telework in Italy with other works identified in the literature

Author Response

We have inserted keywords

We have made the corrections/integrations in the texts and the graphic improvements indicated, expressing the frequency values in the table

We rearticulated the paper in order to better explain in par. 3 the methodological approach assumed

We have rearticulated the literature in order to clarify the specific features of the potentialities and expected impacts of telework in Italy in relation to international experiences

TRANSLATE with x English
Arabic Hebrew Polish
Bulgarian Hindi Portuguese
Catalan Hmong Daw Romanian
Chinese Simplified Hungarian Russian
Chinese Traditional Indonesian Slovak
Czech Italian Slovenian
Danish Japanese Spanish
Dutch Klingon Swedish
English Korean Thai
Estonian Latvian Turkish
Finnish Lithuanian Ukrainian
French Malay Urdu
German Maltese Vietnamese
Greek Norwegian Welsh
Haitian Creole Persian  
TRANSLATE with COPY THE URL BELOW Back EMBED THE SNIPPET BELOW IN YOUR SITE Enable collaborative features and customize widget: Bing Webmaster Portal Back

Reviewer 2 Report

I believe that the manuscript I am reviewing is prepared carefully; however, it seems to be similar to a report rather than a scientific paper. While much information is provided, the theoretical and practical contributions with new knowledge are weak.

I strongly request you re-structure your paper to make clearly sections of data collection, analytical methods, and discussions about the results related to your case study (Italy) and the policy implications proposed.

My specific concerns are as follows:

- The abstract should be written because you provide much information about the background (lines 9-18). The first sentence is very close to the sentence ‘The paper 18 carries out an analysis…’ (line 18). More content of methods, main findings, and implications should be provided there.

- The reference style used does not match the journal guideline.

- Figure 3.1 should be re-drawn rather than a snapshot of a previous paper. And, do you have the right to re-use it?

- The title of Section 3 is ‘3. Teleworking adoption potential in Italy: a study hypothesis’; however, the vast majority of content is about teleworking over the world while little is related to Italy. Please make the general writing about teleworking more concise and better highlight that involved in Italy – your case study.

 - For Table 3.4, I do not know the country ‘Eire’. Are you sure ‘Eire’ is a country? Also in this table, should clarify the context of the study: before or during the pandemic time. Please note that the cases of Italy and Spain are reported by studies carried out in the era of COVID-19. 

- What is the objective of Section 4? Again, this is about a literature review with nothing related to Italy. Please do not overwhelm readers with literature!

- Please define ‘teleworking potential’? how to estimate it? How to get the data to draw Figure 3.3., 4.1?

- The way you cite references is problematic. It seems that you try to cite as many as possible while we need to choose the most informative and updated ones. For example, you cite 11 references of Tremblay, many of which are close together. Reports in French 2001 (Rapports des recherché) can be used to write journal articles in 2002 and 2003. Therefore, you should carefully reconsider references to choose the most appropriate one with much less focus on some particular authors (although they are pioneers of research telework).

- Some state-of-the-art telework-specific studies, you neglected, should be added, as follows:  10.1016/j.tbs.2021.11.003; 

10.1007/s11116-021-10169-5;

10.1016/j.tra.2021.03.027;

10.3390/su13063179

- Some minor points: you should check the manuscript to eliminate grammatical errors (e.g., ‘The sixth and last paragraph is devoted…’ line 57; ‘don’t’ line 232; ‘Mokhtarian (1991) suggest…’ line 235). Some abbreviations are used without explanations (e.g., ICT line 45).

Author Response

In restructuring the paper we have revised the literature references in order to thematise them and distinguish theoretical issues and empirical analysis on the case study of the Italian territory.

We have assumed and integrated in the reflections the recent literature references suggested.

We have reorganised the paper to make explicit in section 3 the methodological approach on the one hand and the references assumed for the empirical analysis of the Italian case study on the other hand

We have made the corrections in the texts and the indicated graphical improvements

TRANSLATE with x English
Arabic Hebrew Polish
Bulgarian Hindi Portuguese
Catalan Hmong Daw Romanian
Chinese Simplified Hungarian Russian
Chinese Traditional Indonesian Slovak
Czech Italian Slovenian
Danish Japanese Spanish
Dutch Klingon Swedish
English Korean Thai
Estonian Latvian Turkish
Finnish Lithuanian Ukrainian
French Malay Urdu
German Maltese Vietnamese
Greek Norwegian Welsh
Haitian Creole Persian  
TRANSLATE with COPY THE URL BELOW Back EMBED THE SNIPPET BELOW IN YOUR SITE Enable collaborative features and customize widget: Bing Webmaster Portal Back

Round 2

Reviewer 1 Report

The authors rectified the problems identified in the previous review and clarified the methodological approach, which resulted in a significant improvement in the article.

Author Response

We have revised the paper for some refinements and to respond to requests from other reviewers.

TRANSLATE with x English
Arabic Hebrew Polish
Bulgarian Hindi Portuguese
Catalan Hmong Daw Romanian
Chinese Simplified Hungarian Russian
Chinese Traditional Indonesian Slovak
Czech Italian Slovenian
Danish Japanese Spanish
Dutch Klingon Swedish
English Korean Thai
Estonian Latvian Turkish
Finnish Lithuanian Ukrainian
French Malay Urdu
German Maltese Vietnamese
Greek Norwegian Welsh
Haitian Creole Persian  
TRANSLATE with COPY THE URL BELOW Back EMBED THE SNIPPET BELOW IN YOUR SITE Enable collaborative features and customize widget: Bing Webmaster Portal Back

Reviewer 2 Report

While I see your several revisions in accordance with my comments; I am dissatisfied with this version because you did not provide detailed responses to my previous comments. As a reviewer, I am responsible for evaluating how you improve your manuscript rather than trying to look at those in your manuscript without a detailed explanation from the authors. Many points I had raised were neglected by you without reason. If you cannot give better and more detailed responses to me; I cannot recommend an acceptance for your work. In your paper of this round, please carefully check tables and figures to remove errors left. Please make your text more concise with more focus on your case study. Please do not abuse references.

Author Response

 We have prepared a table, which we enclose, to illustrate in an orderly manner the work done on the article.

TRANSLATE with x English
Arabic Hebrew Polish
Bulgarian Hindi Portuguese
Catalan Hmong Daw Romanian
Chinese Simplified Hungarian Russian
Chinese Traditional Indonesian Slovak
Czech Italian Slovenian
Danish Japanese Spanish
Dutch Klingon Swedish
English Korean Thai
Estonian Latvian Turkish
Finnish Lithuanian Ukrainian
French Malay Urdu
German Maltese Vietnamese
Greek Norwegian Welsh
Haitian Creole Persian  
TRANSLATE with COPY THE URL BELOW Back EMBED THE SNIPPET BELOW IN YOUR SITE Enable collaborative features and customize widget: Bing Webmaster Portal Back

Round 3

Reviewer 2 Report

Dear the authors,

Thank you for your revisions in accordance with my comments. I am pleased to recommend an acceptance for your work.

Regards,

The reviewer.